# In Situ Observations Reveal the Five-fold Twin-Involved Growth of Gold Nanorods by Particle Attachment

**DOI:** 10.3390/nano13050796

**Published:** 2023-02-21

**Authors:** Qi Sun, Loukya Boddapati, Linan Wang, Junjie Li, Francis Leonard Deepak

**Affiliations:** 1School of Semiconductor Science and Technology, South China Normal University, Foshan 528225, China; 2Research Center for Crystal Materials, CAS Key Laboratory of Functional Materials and Devices for Special Environments, Xinjiang Technical Institute of Physics & Chemistry, CAS, Urumqi 830011, China; 3Nanostructured Materials Group, International Iberian Nanotechnology Laboratory (INL), Avenida Mestre Jose Veiga, 4715-330 Braga, Portugal; 4Center of Materials Science and Optoelectronics Engineering, University of Chinese Academy of Sciences, Beijing 100049, China

**Keywords:** in situ observation, Au nanorod, crystal growth, particle attachment, transmission electron microscopy

## Abstract

Crystallization plays a critical role in determining crystal size, purity and morphology. Therefore, uncovering the growth dynamics of nanoparticles (NPs) atomically is important for the controllable fabrication of nanocrystals with desired geometry and properties. Herein, we conducted in situ atomic-scale observations on the growth of Au nanorods (NRs) by particle attachment within an aberration-corrected transmission electron microscope (AC-TEM). The results show that the attachment of spherical colloidal Au NPs with a size of about 10 nm involves the formation and growth of neck-like (NL) structures, followed by five-fold twin intermediate states and total atomic rearrangement. The statistical analyses show that the length and diameter of Au NRs can be well regulated by the number of tip-to-tip Au NPs and the size of colloidal Au NPs, respectively. The results highlight five-fold twin-involved particle attachment in spherical Au NPs with a size of 3–14 nm, and provide insights into the fabrication of Au NRs using irradiation chemistry.

## 1. Introduction

Geometrical structure and chemical composition are the two important parameters that have a critical influence on the properties of functional materials [1,2,3,4]. NPs with zero-/one-/two-dimensional (0D/1D/2D) structures have received considerable attention in recent decades, and the fabrication of all kinds of nanostructures has been of central interest in this field [5,6,7,8,9,10,11,12,13,14,15,16,17]. Gold (Au) NPs, as one of the promising candidates for catalytic, electronic, plasmonic, molecular diagnostic and sensing applications, are well known for their tailorable size and surface chemistry, high stability and good biocompatibility [18,19,20,21,22,23]. In 1857, Faraday reported the synthesis of colloidal Au NPs for the first time [24]. Thereafter, a large number of experiments have been carried out to fabricate Au NPs with controllable size and unique morphology, involving various routes such as wet chemical synthesis at room temperature, hydrothermal processes, template methods and irradiation-assisted routes, all of which have been developed (for the synthesis of Au NPs) over recent decades [25,26]. Meanwhile, in order to clarify the crystal growth mechanism and thus achieve controllable chemical synthesis of Au NPs, the crystal growth dynamics of Au NPs have been investigated by ex situ and/or in situ observation techniques such as X-ray scattering (XRS) techniques, UV-VIS spectroscopy, X-ray absorption fine structure (XAFS) spectroscopy, atomic force microscopy (AFM), scanning electron microscopy (SEM), transmission electron microscopy (TEM), scanning TEM (STEM) and so on [21,27,28,29,30,31]. Among them, microscopies, especially liquid cell TEM (LCTEM), have been regarded as a powerful tool to visualize crystal growth kinetics at the nano- or atomic-scale [32,33,34]. 

Based on classical crystal growth models, crystals are grown by simple structure replication, which does not involve structural transition or intermediate states. However, the classical mechanism or model cannot explain all of the phenomena that have recently been discovered in many crystallization processes. Ex situ and in situ studies have uncovered the existence of nonclassical crystal growth processes, including intraparticle growth [35,36], oriented attachment (OA) growth [37], grain boundary migration and surface-atom-migration dominated coalescence and growth pathways [38,39]. For examples, Tang et al. investigated the growth of Au NPs in 10.0 mM HAuCl_4_ solutions by in situ LCTEM [40]. The formation of “cluster-cloud” (condensed atomic clusters) was observed in the crystallization, which plays a critical role in the formation of the initial crystal nucleus and crystal growth [40]. To illustrate the formation of a superlattice, Chen and Luijten et al. investigated the growth of Au superlattices from Au nanoprisms by in situ LCTEM in combination with Monte Carlo simulations [41]. To clarify the influence of ligands, the ligand-controlled OA processes were studied by Sun et al. in a LCTEM at the atomic scale [42]. To detect the substrate-effected coalescence growth, the coalescence process of Au NPs on crystalline (MgO) and amorphous (carbon) supports were studied by Li and Deepak et al. [21,23]. The results demonstrate that, compared to the amorphous carbon substrate, the crystalline MgO support is suitable for the migration of Au NPs/atoms, giving rise to rapid crystal growth kinetics. To uncover the influence of concentration, Kim et al. investigated the formation of spiky Au NPs from HAuCl_4_ solutions with a broad solution concentration range by combining UV-vis spectra and in situ TEM, demonstrating the HAuCl_4_ concentration-dependent crystal growth processes [43]. More recently, Li et al. uncovered the formation of Au NPs with a five-fold twinned structure by in situ TEM in combination with molecular dynamic simulations at the atomic scale [44]. The in situ dynamic observations provide nano- or atomic-scale information towards understanding the classical or non-classical crystal growth processes. Nevertheless, most of the in situ dynamic investigations were focused on the growth of Au NPs with a length/diameter aspect ratio of ∼1, and studies on the growth of 1D Au NRs or NWs were quite limited [45,46,47].

In this work, we report in situ atomic-scale dynamic observations on the growth of Au NRs in an aberration-corrected TEM (ACTEM). To obtain atomic-scale resolution, highly dispersed Au NPs ~10 nm were employed for the in situ experiments. The results show that the Au NRs with length/diameter aspect ratios of ~1–3 are formed by the particle attachment mechanism under electron irradiation. At the initial stage, the Au NPs are connected with each other in a tip-to-tip way to construct the NL structures, which is different to the formation of a super-lattice by the direct particle attachment of cubic NPs. Subsequently, the formed NL structures show linear growth dynamics to achieve closed height and width with the adjacent NPs. Finally, the Au NPs display a total atomic rearrangement to construct a Au NR with low surface energy. It is worth noting that the five-fold twin structures are involved in the atomic arrangement process because of their low formation energy in the Au NPs with the size of ~3 to 14 nm. Moreover, the statistical analyses indicate that the length of the Au NRs can be easily controlled by the number of attached Au NPs, and their diameter is mainly dependent on the size of the attached Au NPs.

## 2. Experimental Section

### 2.1. Sample Preparation and Characterization

The highly dispersed citrate-capped Au NPs (size: 6 nm to 10 nm) in H_2_O were obtained from BBI Solutions Co., Ltd. Wales, UK (Gold Colloid (bbisolutions.com), 30 June 2020). To avoid possible aggregation, the colloidal Au NPs were stored in the original solution and used without further purification. Before utilization, the Au samples were diluted and dispersed by ultrasonication, and a few drops of this dispersion of Au solution were dropped onto a carbon grid and then dried for structural characterizations and in situ TEM observations.

### 2.2. In situ TEM Observations

TEM imaging was carried out at 200 kV using a Titan Themis TEM equipped with both an image Cs corrector and probe Cs corrector, which provided an opportunity to probe the dynamic process under sub-angstrom resolution [19,48,49,50]. The images or videos were acquired using Tecnai Imaging and Analysis (TIA) software. The electron dose rates were controlled by the spot size.

## 3. Results and Discussion

To reduce the influence of size, the commercial Au NPs with a diameter of ~ 10 nm were used for the dynamic observations. Before the experiments, the colloidal Au samples were dispersed by ultrasonication. Figure 1a shows the sphere-like morphology of the Au NPs used. Based on the statistical analyses on the diameter of 22 Au NPs in different regions in different TEM images, it can be seen that the colloidal Au NPs show a size distribution from 6 nm to 11 nm, and the averaged size is about 9 nm, as shown in Figure 1b. To confirm the crystal structure, high-resolution TEM images were acquired. Figure 1c shows a high resolution TEM image of a Au NP, and Figure 1d is the corresponding Fourier transformation (FFT) analysis. The results indicate the face-centered cubic structure in the colloidal Au NPs.

To probe the growth dynamic process of Au NRs, both STEM and TEM modes were employed, as shown in Figure 2. In Figure 2a–c, the sequential high-angle annular dark field (HAADF)-STEM images show that the Au NR can be grown by a particle attachment mechanism. However, in contrast with the direct particle attachment of cubic NPs in the formation of super-lattices of MOF, the spherical NP attachment involved the coalescence of a round-like grain boundary and total atomic rearrangement to minimize the surface energy and total energy. As shown in Figure 2b, a NL structure was formed at the initial stage. Subsequently, the NL structure grew to form a dumbbell-like structure (Figure 2c). Due to the disturbance of the scanning electron probe on the surface structure, as well as the carbon contamination that starts to build-up during in situ observations, it was challenging to obtain high-quality videos under the HAADF-STEM mode. 

The attachment of two Au NPs (7.2 nm and 8.5 nm) were further recorded in TEM mode under an electron dose rate of 1.0 × 10^5^ e/Å ^2^s. Figure 2d–i show the sequential atomic scale TEM results, which were obtained from Appendix A. Similar to the HAADF-STEM results, the NL structure (Figure 2d) was further confirmed under TEM mode. Thereafter, the two Au NPs showed total atomic rearrangement, in which the five-fold twin structures were involved, as shown in Figure 2e–h and Appendix A. The five-fold twin structure in the upper Au NP appeared at the initial contacting stage of the two Au NPs (Figure 2d-e). Thereafter, under the drive of minimizing the surface energy and total energy, the upper five-fold twin structure and the lower Au nanocrystal (Figure 2h) showed a total atomic rearrangement to achieve a closed or a same orientation (ideally) by forming smooth surface and reducing the interior defects like the twin boundaries in the NR. Moreover, the formed five-fold twin structure could be related to the size effect. Based on the thermodynamic analyses, the five-fold twin structure of Au NPs is thermodynamically stable with the size of 3 to 14 nm [44,51,52,53]. When the size of Au NP is larger than 14 nm, the five-fold twin structure disappears, as shown in Figure 2i. At 582 s, the mean diameter for the formed Au NR was measured to be ~8.5 nm and the ratio of length/diameter was about 2.1.

To further check the growth dynamics of the NL structure, the change of the “neck” diameter along with time was investigated. Based on the sequential high-resolution TEM images in Appendix A, the statistical analyses were carried out, and the results indicated a linear growth dynamic for the NL structure, as shown in Figure 3, which should be related to the atomic migration rate.

The schematic diagram shown below (Figure 4) illustrates the attachment of spherical Au NPs with a size of 3 to 14 nm. As shown in Figure 4, the separated Au NPs first connected with each other to form an NL structure, and then showed a linear growth and total atomic rearrangement involving the formation and disappearance of the five-fold twin structure. Driven by minimizing the surface energy and total energy, the Au NR could be formed.

To detect the length-dependent-growth dynamics, three and four Au NP attachment (in a tip-to-tip way)-driven crystal growth of Au NRs were carried out, respectively. Figure 5a–f (Appendix A) show the formation of a Au NR (Figure 5f) with a length of ~19 nm by grouping three Au NPs (~8.9 nm, ~9.4 nm and ~8.4 nm) under an electron dose rate of ~1.0 × 10^5^ e/Å ^2^s. Figure 5g–l (Appendix A) show the formation of a Au NR (Figure 5l) with a length of ~32.5 nm by attaching four Au NPs (~9.8 nm, ~9.0 nm, ~8.1 nm and ~9.8 nm) in a tip-to-tip way under an electron dose rate of ~1.0 × 10^5^ e/Å ^2^s. Similar to the attachment of two Au NPs in Figure 2, the attachment of three/four spherical Au NPs also shows the NL structures when the Au NPs are initially contacted with each other, and then follows a total atomic rearrangement and grain boundary migration to achieve a rod-like geometry from the bead-like structures. The diameters for the two NRs were 8.4 nm (Figure 5f) and 8.2 nm (averaged value in Figure 5l), and the length/diameter were ~2.3 (Figure 5f) and ~3.8 (Figure 5l), respectively. It means that the length is dependent on the number of attached Au NPs. Moreover, the thermodynamically stable five-fold twin structure in the small Au NPs with a size between 3 to 14 nm was also observed in both in situ dynamic observations (Appendix A).

To shed further light on the structural evolution and to quantify the growth dynamics of the Au NRs with different lengths in Appendix A, the change of length (l), diameter (d) and length–diameter ratio (l/d) with time were investigated by statistical analyses. It is worth noting that the diameter d is an average value here. The results show that the length of the NPs is reduced with time (Figure 6a–c), and the diameter is related to the size of spherical Au NPs and time (Figure 6d–f). At the initial stage, the values of length and averaged width fluctuated rapidly, which can be attributed to the formation and growth of NL structures. The aspect (length/diameter) ratios (Figure 6g–i) indicate that the length of the Au NR formed via particle attachment is mainly determined by the number of tip-to-tip Au NPs, and the aspect (length/diameter) ratio can be modified by regulating the attachment time.

## 4. Conclusions

The in situ atomic-scale observations on the formation of Au NRs by spherical colloidal Au NPs represents a significant step forward in understanding the growth mechanisms of NRs using spherical particle attachment. The experimental results show that the attachment of spherical Au NPs includes the formation and growth of NL structures and the atomic rearrangement involves the formation of five-fold twin structures in Au NPs with a size of ~3 to 14 nm. Moreover, the statistical analyses indicate that the formed NL structure displayed linear growth dynamics. The length of Au NRs can be well regulated by the number of tip-to-tip Au NPs, and the diameters are mainly determined by the size of spherical Au NPs. These findings indicate that the attachment of spherical NPs during the growth of Au NRs could be different to the attachment of cubic NPs in the formation of a super-lattice, and provide new insights into the fabrication of Au NRs based on colloidal Au NPs and irradiation chemistry.

## Figures and Tables

**Figure 1 nanomaterials-13-00796-f001:**
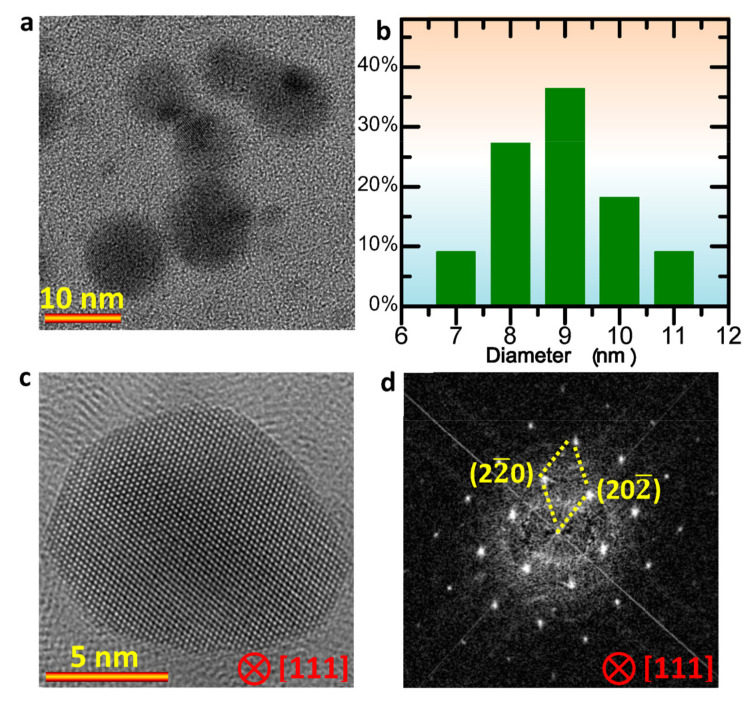
The morphology and size of Au NPs. (**a**) TEM image showing the morphology of Au NPs; (**b**) statistical analysis showing the size distribution of the highly dispersed Au NPs; (**c**,**d**) high-resolution TEM image and corresponding FFT pattern of a Au NP.

**Figure 2 nanomaterials-13-00796-f002:**
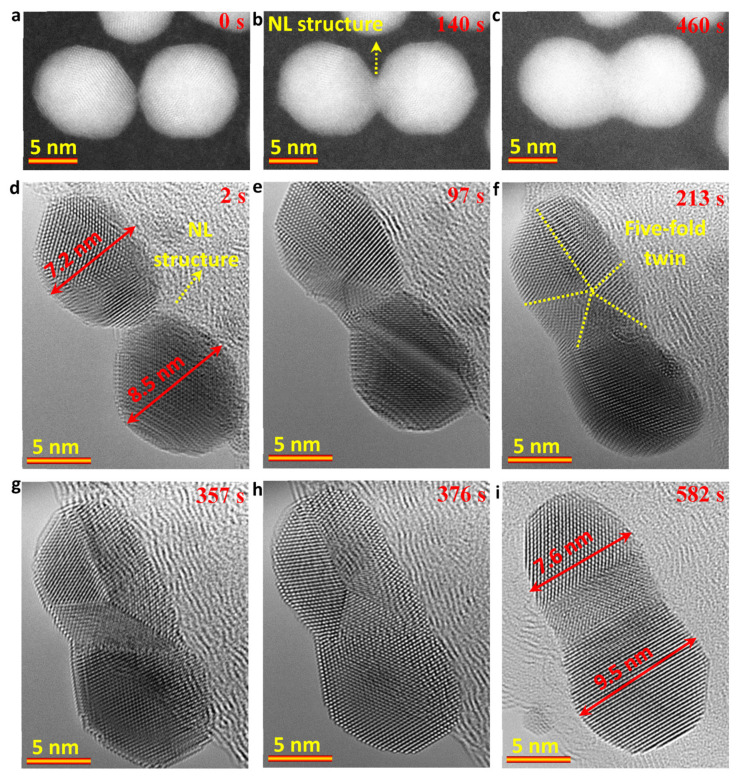
(**a**–**c**) Sequential HAADF-STEM images showing the growth of Au NRs with a length of ~20 nm from two Au NPs; (**d**–**i**) Sequential TEM images from Appendix A showing the growth of Au NRs with a length of ~18 nm at atomic scale by the attachment of two Au NPs. The electron dose rate is ~1.0 × 10^5^ e/Å^2^s.

**Figure 3 nanomaterials-13-00796-f003:**
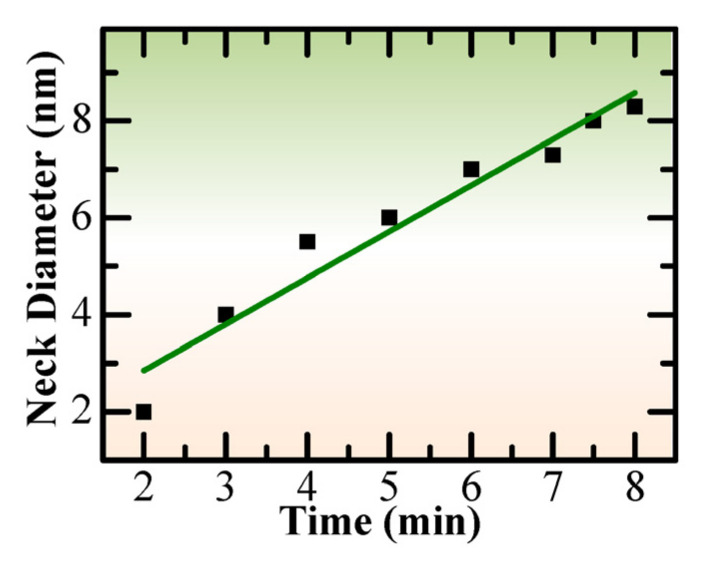
Statistical analysis showing the linear growth of neck diameter with time in Appendix A.

**Figure 4 nanomaterials-13-00796-f004:**
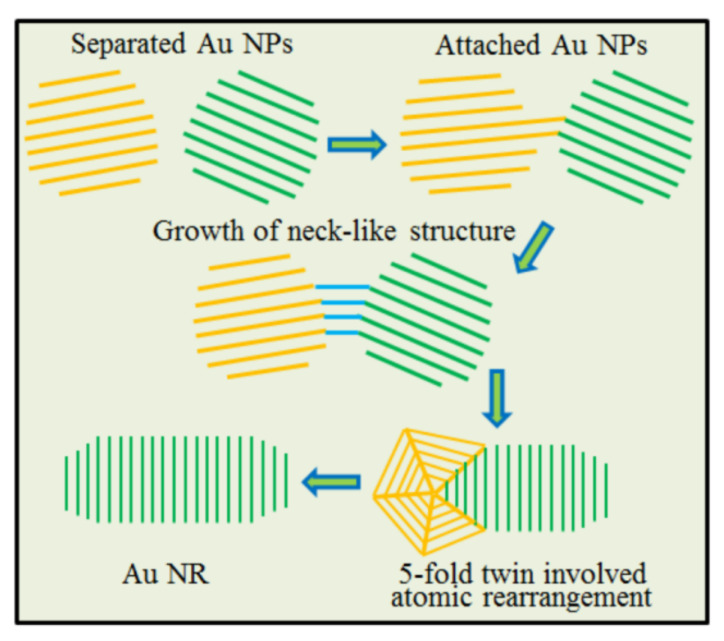
Schematic diagram showing the formation of Au NR by the attachment of Au NPs with a size of ~10 nm.

**Figure 5 nanomaterials-13-00796-f005:**
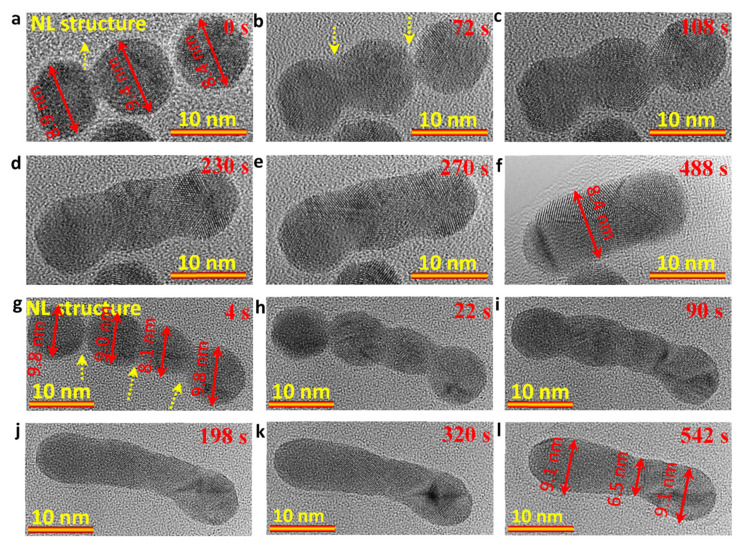
(**a**–**f**) Sequential TEM images from Appendix A showing the growth of Au NRs with a length of ~19 nm through the attachment of three Au NPs in a tip-to-tip way; (**g**–**l**) Sequential TEM images from Appendix A showing the growth of Au NRs with a length of ~32.5 nm through the attachment of four Au NPs in a tip-to-tip way. The electron dose rate is ~1.0 × 10^5^ e/Å^2^s.

**Figure 6 nanomaterials-13-00796-f006:**
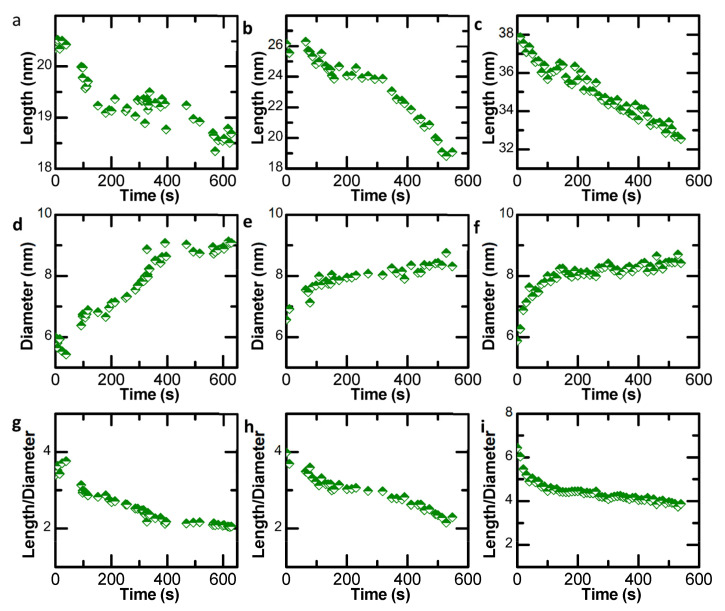
Statistical analyses showing the fluctuation of length (**a**–**c**), diameter (**d**–**f**) and aspect (length/diameter) ratio (**g**–**i**) of Au NRs along with time in Appendix A. Appendix A were played at seven times that of normal speed.

## Data Availability

The data presented in this study are available on request from the corresponding author.

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
