# Peer review of "In Situ Observations Reveal the Five-fold Twin-Involved Growth of Gold Nanorods by Particle Attachment"

_nanomaterials, 2023, doi:10.3390/nano13050796_

Round 1

Reviewer 1 Report

The authors present in situ atomic-scale observations on the formation of Au nanorods by attaching of spherical colloidal Au nanoparticles. The TEM investigation have revealed the growth kinetics of nanorods. Intermediate necking-like structures were shaped after coalescence of Au particles.  Subsequently, a 5-fold twin structure was formed after atomic rearrangement.

The paper has shown importance of in situ TEM studies for revealing the mechanism of nanorod formation. It is clearly written and needs only a few technical corrections:

(1)    Please, correct units on p. 4 line 26 and p. 6 line 20: (e/A2s): A => Å

(2)    Please, correct on p. 4, line 7: shonw => shown

(3)    Fragments of the lettering of some figures (1, 5) appear randomly in the text at least in my PDF copy.  Please, check this fact and correct the fault.

Author Response

We would like to acknowledge the reviewers very much for reviewing our manuscript very carefully and providing helpful comments. The revised opinions have greatly improved the manuscript. We have revised the manuscript thoroughly taking fully these comments into account. Changes; related to the reviewers’ comments, are indicated in red in the response text, revised text and supporting information. Our responses to specific comments are given below.

Response to Referee 1

The authors present in situ atomic-scale observations on the formation of Au nanorods by attaching of spherical colloidal Au nanoparticles. The TEM investigation have revealed the growth kinetics of nanorods. Intermediate necking-like structures were shaped after coalescence of Au particles.  Subsequently, a 5-fold twin structure was formed after atomic rearrangement.

The paper has shown importance of in situ TEM studies for revealing the mechanism of nanorod formation. It is clearly written and needs only a few technical corrections:

  • Please, correct units on p. 4 line 26 and p. 6 line 20: (e/A2s): A => Å

Response: We thank the reviewer for this comment. The units have been corrected to e/Å2s in the revised manuscript.

  • Please, correct on p. 4, line 7: shonw => shown

Response: We thank the reviewer for this comment. The typo has been corrected in the revised manuscript.

  • Fragments of the lettering of some figures (1, 5) appear randomly in the text at least in my PDF copy.  Please, check this fact and correct the fault.

Response: We thank the reviewer for this comment. The fragments have been grouped in the Figures.

Reviewer 2 Report

    This article is a very short experimental paper in which the fusion of two crystalline gold nanoparticles is investigated using transmission electron microscopy. The study of the fusion process of two crystallites is very important for many areas of crystal physics and engineering. Therefore, I believe that this work is very relevant. In this case it was found that the fusion occurs through the formation of an intermediate crystal structure with a 5-fold twin structure. This is a beautiful result, but immediately there is a question that wants to discuss in the paper. 

    The authors believe that this 5-fold twin structure is only characteristic of gold or similar structures should be observed in the coagulation of crystalline nanoparticles from other materials with the same cubic symmetry group Fm-3m, such as Cu, Ag, Al, etc.? Understanding this question would be important for other areas of crystal growth.

Author Response

We would like to acknowledge the reviewers very much for reviewing our manuscript very carefully and providing helpful comments. The revised opinions have greatly improved the manuscript. We have revised the manuscript thoroughly taking fully these comments into account. Changes; related to the reviewers’ comments, are indicated in red in the response text, revised text and supporting information. Our responses to specific comments are given below.

Response to Referee 2

This article is a very short experimental paper in which the fusion of two crystalline gold nanoparticles is investigated using transmission electron microscopy. The study of the fusion process of two crystallites is very important for many areas of crystal physics and engineering. Therefore, I believe that this work is very relevant. In this case it was found that the fusion occurs through the formation of an intermediate crystal structure with a 5-fold twin structure. This is a beautiful result, but immediately there is a question that wants to discuss in the paper. 

    The authors believe that this 5-fold twin structure is only characteristic of gold or similar structures should be observed in the coagulation of crystalline nanoparticles from other materials with the same cubic symmetry group Fm-3m, such as Cu, Ag, Al, etc.? Understanding this question would be important for other areas of crystal growth.

Response: We thank the reviewer for this comment. The observed 5-fold twin intermediate structure is related to their formation energy. If the formation energy is lower than or close to the one in a single crystal, the 5-fold twin structure could be thermodynamically stable, which is usually observed in the face-centered cubic metals at the nanoscale.1-3 Correspondingly, the discussions “Moreover, the formed five-fold twin structure could be related to the size effect. Based on the thermodynamic analyses, the five-fold twin structure of Au NPs is thermodynamically stable with the size of 3 to 14 nm.” have been added in the revised manuscript on page 5 lines 9-12.

References:

  1. Barnard, A. S.; Young, N. P.; Kirkland, A. I.; van Huis, M. A.; Xu, H, F. Nanogold: a quantitative phase map. ACS nano 2009, 3, (6), 1431-1436.
  2. Koga, K.; Ikeshoji, T.; Sugawara, K. Size- and temperature-dependent structural transitions in gold nanoparticles. Physical review letters 2004, 92, (11), 115507.
  3. Young, N. P.; van Huis, M. A.; Zandbergen, H. W.; Xu, H.; Kirkland, A. I. Transformations of gold nanoparticles investigated using variable temperature high-resolution transmission electron microscopy. Ultramicroscopy 2010, 110, (5), 506-516.
